# UK Doctors Delivering Physical Activity Advice: What Are the Challenges and Possible Solutions? A Qualitative Study

**DOI:** 10.3390/ijerph191912030

**Published:** 2022-09-23

**Authors:** Dane Vishnubala, Adil Iqbal, Katherine Marino, Steven Whatmough, Ruth Barker, David Salman, Peter Bazira, Gabrielle Finn, Andy Pringle, Camilla Nykjaer

**Affiliations:** 1Health Professions Education Unit, Hull York Medical School, York YO10 5DD, UK; 2Faculty of Biological Sciences, University of Leeds, Leeds LS2 9JT, UK; 3School of Public Health, Imperial College London, London SW7 2BX, UK; 4MSK Lab, Imperial College London, London SW7 2BX, UK; 5Bradford Teaching Hospital Foundation Trust, Bradford BD9 6RJ, UK; 6Royal Stoke University Hospital, Stoke-on-Trent ST4 6QG, UK; 7Leeds Teaching Hospital, Leeds LS7 4SA, UK; 8Faculty of Biology, Medicine and Health, University of Manchester, Manchester M13 9PL, UK; 9Sport Outdoor and Exercise Science, School of Human Sciences, Human Science Research Centre University of Derby, Derby DE22 1GB, UK

**Keywords:** physical activity, adults, medical education

## Abstract

Despite strategies to enable Health Care Professionals (HCPs) to give physical activity (PA) advice to patients, this appears to be rarely done in consultations. The aims of the present study were to gain an understanding of doctors’ awareness of current PA guidelines and to explore their opinions on barriers and solutions. A qualitative approach using semi-structured interviews was adopted. This study included 15 doctors currently working in the UK’s National Health Service (NHS). A thematic analysis approach was used to analyse the transcripts. Four themes and twelve sub-themes were deciphered. Intrinsic factors limiting the delivery of PA advice included a lack of knowledge of PA guidelines and PA being an afterthought. Barriers to delivering PA guidance included a lack of PA education, time pressures, and patient engagement. Solutions included staff training, incorporating PA into undergraduate training, and encouraging staff to be physically active. Methods to optimise PA guidance included individualised PA advice, local exercise services and schemes, utilising online and visual resources, and motivational interviewing. This study provides an updated insight into doctors’ opinions on barriers and solutions to discussing PA with patients. It is clear that further work is needed to ensure greater awareness of PA guidelines amongst clinicians.

## 1. Introduction

Despite significant evidence that physical activity (PA) has numerous benefits on physical and mental health and well-being, levels of physical inactivity are increasing [1,2]. In the United Kingdom (UK), the Chief Medical Officer (CMO) recommends that adults aged 19–64 years should complete at least 150 min of moderate activity, or 75 min of vigorous activity, each week, as well as muscle-strengthening activities at least twice per week [3]. Despite the well-documented benefits of PA, most adults globally fail to reach these recommendations [2]. According to the most recent Active Lives Adult Survey, between November 2020 and 2021 in the UK, 61% of adults were receiving at least 150 min of PA per week, with 27% classed as inactive, meaning they engaged in less than 30 min of moderate PA per week [4]. In recent years, physical inactivity and sedentary behaviour have been under increased focus as they have been found to be independent risk factors for mortality and non-communicable diseases [5,6]. With the COVID-19 pandemic and subsequent lockdowns, researchers have found that PA levels in the UK have decreased further, associated with reduced physical function and increased mental ill-health [7,8]. It is, therefore, of increasing importance to optimise strategies to increase engagement with PA.

The UK CMO highlighted the importance of healthcare professionals (HCPs) in promoting PA to patients in their updated guidance released in 2019 [3]. There have been schemes to engage HCPs in discussing PA with patients, such as Make Every Contact Count (MECC), which aims to improve confident discussions of PA with patients through HCP training [9]. Despite this, a recent study of 839 UK-based GPs found that only 36% were “somewhat familiar” with the PA guidelines [10].

The aims of the present study were to gain an updated, in-depth understanding of doctors’ awareness of current PA guidelines and explore their opinions of any potential barriers and solutions for doctors engaging in PA discussions with their patients.

## 2. Materials and Methods

### 2.1. Design

A qualitative approach was taken, utilising semi-structured interviews [11]. Ethics approval was given by the Faculty of Biological Sciences at the University of Leeds (27 July 2020/BIOSCI 19-039).

### 2.2. Participants and Procedures

The inclusion criteria for this study included UK-based doctors currently working in the National Health Service (NHS), including doctors working in hospitals and primary care. Participants were excluded from the study if their medical degree was not from a UK University or if they were currently not practising medicine. The participant recruitment process for this study involved advertising through a range of channels, including social media (Twitter and LinkedIn) and by word of mouth. Recruitment was aimed at clinicians who are patient-facing but working across primary and secondary care. This was intentionally broad to reflect the opinions of all those participants who would normally be expected to provide PA guidance. Guidance on PA concerns almost all clinicians working within healthcare. Participants that expressed interest in these channels were sent a copy of the participant information sheet and consent forms, which were signed electronically and returned. Participants were contacted to arrange an interview either in person or online via Zoom. At the start of each interview, participants confirmed they had read the participant information sheet and gave verbal consent to be included in the study. Recruitment and interviews continued until data saturation was reached; this is the point at which no new emerging themes were identified [12].

### 2.3. Data Collection

Data were collected using semi-structured interviews to gather in-depth and informative responses [13]. This interview method was chosen as the research design as it enabled the interviewer to clarify statements and enquire about further information [11]. The interview guide contained 20 questions and is provided in the Appendix A. The questions promoted open dialogue between the interviewer and participant to elicit maximal information [13], and probing was used when appropriate if initial responses were limited [14]. The last question of the interview enquired whether the participants had any other comments to make to allow for new information and ideas [15]. Several questions were specifically included in the interview to gain an understanding of the participant’s awareness of PA guidelines, including asking if they meet the current UK guidelines for PA themselves, awareness of the PA guidelines, awareness of the CMO PA guidance and awareness of the Moving Medicine resource, an online PA resource for HCPs (https://movingmedicine.ac.uk/).

DV, AI, RB, and SW were all involved in data collection. DV provided training to AI, RB, and SW and observed one interview per interviewer to ensure consistency. Interviews were conducted via Zoom between March 2021 and May 2021; only audio was recorded for all interviews. Any identifiable information on the recordings was removed. It was made clear that data would remain anonymous and confidential, and participation was entirely optional. Participants were informed that they could withdraw from the interview at any stage. Unique codes were used throughout the study to preserve participants’ identities. Recordings were transcribed verbatim.

### 2.4. Data Analysis

Thematic analysis was used to analyse the transcripts [16], which enables a rich and detailed account of the data to be obtained [17]. The six-step process is used to extract meaning and concepts from data to identify patterns and ultimately generate themes [18]. Refinements to the themes and subthemes continued until nothing substantial was added. A recursive process was used in the analysis, moving back and forth between transcripts and themes as needed [16]. Themes were reviewed regularly until there was a distinct and coherent meaning for each. Transcripts were analysed by KM and DV separately. A reflective journal was utilised to evidence the extent to which thoughts and observations were data-driven and without researcher influence, in turn reducing the likelihood of researcher bias. Together, themes and sub-themes were discussed and agreed upon. NVIVO 11 software was utilised to manage extracts from the interviews and illustrate themes. Example quotes from transcripts were presented in tables for each theme and sub-theme. Signs of data saturation were first seen at 10 participants, and no new themes were identified following participants 12 to 15, signifying complete data saturation and that enough information was gained to ensure repeatability [19]. Microsoft Excel was used to collate and summarise qualitative data, which included demographic information of participants (years of experience, current primary healthcare setting, job role and location of work), and frequencies and proportions of responses to questions asked about participant understanding and awareness of PA guidelines.

## 3. Results

### 3.1. Participant Characteristics

Fifteen participants expressed interest in participating in the study. As all were eligible, a total of 15 participants were included in the study. Interviews varied in length between 20 min and 45 min, with an average of 29 min. Data saturation was reached at participant 15; at this point, no more emerging themes or new responses were found. Due to data saturation being reached, no further participants were recruited.

Participant characteristics can be found in Table 1 below. The majority of participants (73%) had 6–10 years of experience working as a doctor. The job role of participants varied widely across different specialities, from foundation year 1 doctors to qualified General Practitioners and one Sport and Exercise Medicine Consultant. The majority of participants were located in either northwest England (60%) or Yorkshire and Humber (27%). While the majority of participants were physically active and meeting the CMO PA guidelines (80%), only 27% of participants were aware of the CMO PA Guidelines, and only 33% were aware of Moving Medicine as a resource.

### 3.2. Themes and Sub-Themes

A total of 122 codes were generated from the 15 transcripts. Connections between these codes were found to coalesce into 4 themes and 12 sub-themes, summarised in Table 2. Themes and subthemes are listed in order of when they were deciphered from the data; the order does not signify importance. Next to the participant quote is the assigned identification number for the participant in brackets.

### 3.3. Theme 1—Intrinsic Factors Limiting Delivery of PA Advice

#### 3.3.1. Lack of Knowledge of PA Guidelines

A lack of knowledge and awareness of the PA guidelines was mentioned in the majority of interviews. There was a broad lack of awareness of PA guidelines amongst the interviewees. Similar comments were made by several interviewees when asked about their knowledge of PA, with one stating: ‘I wouldn’t say I [I am] confident to be honest’ (2) and another commenting: ‘I think I’ve looked at them many years ago, but they don’t come to mind particularly often’ (6). Going a step beyond the knowledge of PA guidelines to giving advice and prescribing exercise, one interviewee stated: ‘I feel confident in prescribing medication. I feel less confident in giving exercise advice’ (15).

#### 3.3.2. PA Is Often an Afterthought

Many interviewees emphasised that PA is often not a top priority for medical professionals during consultations, with one interviewee stating: ‘It’s tricky because it’s often not the top thing that you talk about’ (11). Several interviewees asserted that it would be advantageous to incorporate PA into the taught consultation structure: ‘If it becomes part of our natural history taking and we’re looking at them as a whole… you could then capture a lot of factors that can then change in the future’ (14).

### 3.4. Theme 2—Barriers to Delivering PA Guidance

#### 3.4.1. Lack of PA Education

A lack of PA education during undergraduate and postgraduate training was highlighted during the interviews. One interviewee stated when asked if they had received any PA education: ‘Nothing formal from my recollection’ (9), and another stated ‘I wouldn’t say I received formal education with regards to delivering physical activity advice. It’s been a more informal and implicit kind of education mostly during my time on placement during medical school’ (1). Another interviewee added to this: ‘a lot of it does stem from the neglect from medical school regarding physical activity teaching’ (3).

#### 3.4.2. Time Pressures

Time constraints were mentioned in most interviews and by all those working in primary care settings. Regarding giving PA advice in practice, one interviewee said: ‘with their time constraints of a GP 10 min consultations you’re less likely to achieve it’ (5). Another added a similar comment: ‘I know GPs are hard-pressed; they’ve got so much pressure to deal with so many issues coming in within 10 min. That is just not possible to give effective PA advice that patients can take in and understand’ (2). However, several interviewees did express altering views, with one interviewee stating: ‘it doesn’t take that much time to add [physical activity] into what you’re speaking about’ (4).

#### 3.4.3. Patient Engagement

It was mentioned in multiple interviews that the patients themselves, understandably, play a large role in whether PA advice will be successful in creating behavioural change. One interviewee explained: ‘It’s very patient-dependent. He wants to lose weight and he’s keen to get [PA advice] it’s easier to talk to them’ (12). Self-motivation was highlighted as an important factor, with the same interviewee stating: ‘it’s more like empowering them to do it themselves as well’ (12). Another added: ‘If they’re not ready to engage, there’s no point in me saying “you should run half an hour, five times a week” because they’re just going to go on deaf ears and I’m going to lose that rapport with them’ (15).

### 3.5. Theme 3—Solutions to Increasing Staff Awareness and Successful Implementation of PA Guidelines

#### 3.5.1. Staff Training

Increased education and teaching of PA for medical staff was a common theme throughout the transcripts. One interviewee stated: ‘I think a baseline would be useful for us all because it’s so beneficial and pretty much for every single patient, especially with chronic disease’ (8). Regarding staff training, the importance of including up-to-date research and statistics in PA education was mentioned by interviewees, with one stating: ‘have those comparison tables, comparison charts that show statistics even if the number needed to treat’ (4). Another interviewee commented: ‘obviously evidence-based medicine is such an important part of delivering safe and high-quality patient care’ (1), and another added: ‘if you empower them, give them the knowledge that it is not just ‘oh it might be useful’ but actually clinically it’s been shown the evidence is there and the evidence is strong’ (5).

#### 3.5.2. Incorporating PA into Undergraduate Training

Incorporating more PA into undergraduate curricula was mentioned by many interviewees as a method of increasing knowledge among medical professionals. One interviewee stated: ‘I think that’s really important to do, yes 100%’ (12). Another interviewee commented: ‘it needs to be better integrated within medical school and linking it in towards the earlier years of your career as well because that’s where you start developing your practice in action’ (1).

#### 3.5.3. Encouraging Staff to Be Active Themselves

The concept of encouraging staff to be physically active was mentioned by several interviewees, with one commenting: ‘I suppose if you’re more active yourself, it’s probably easier to give tailored advice to people who live in the same city as you’(4). Another interviewee stated: ‘it’s a case of practice what you preach’ (5).

### 3.6. Theme 4—Methods to Optimise PA Advice

#### 3.6.1. Individualised PA Advice

Ensuring that PA advice is targeted individually was mentioned numerous times throughout the interviews, highlighting that it is beneficial to have a good knowledge and understanding of the person you are pitching the PA advice to. One interviewee stated: ‘you have to know your patient and know how best to pitch the idea to them’ (12). Another stated: ‘I think finding out what they’re interested in, some people prefer different forms of exercise’ (14), and they added ‘if you can work it into their daily routine, walk to school rather than drive to school sort of things can really make a difference in terms of like the whole family exercise’ (14). Motivational interviewing was highlighted as an important aspect of optimising PA advice. One interviewee stated: ‘I think having that motivational interviewing approach to the consultation has helped, I think bearing in mind physical activity when talking about different chronic or acute health conditions has helped, again, patient-led’ (7). Another interviewee commented on how they have personally found motivational interviewing to be effective: ‘I usually use now some motivational interviewing approach, starting with assessing via open questions, what they do at the moment, what they know about the possible benefits of exercise, whether they’ve contemplated—where they are on the state of change sequence from that point of view’ (10).

#### 3.6.2. Local Exercise Services and Schemes

Local exercise services, such as referring to an exercise programme, and schemes such as the couch-to-5k, were highlighted by several interviewees as useful methods to encourage patients to increase their PA levels. One interviewee said: ‘people seem to really like that flexibility around it’ (15). Another interviewee gave a specific example: ‘one of them is a local football team and they’ve set up half-term fitness regimes and places that kids can go and learn about diet and exercise, so I have referred to them a couple of times’ (14).

#### 3.6.3. Utilising Online and Visual Resources

The use of visual and online resources was highlighted, with one interviewee stating: ‘I like to give outpatient leaflets, especially if they’ve got an arthritic issue, for example, that has exercises on there that they can use’ (15), and adding to this ‘in recent time, you can text people with the website, so I’ll text them with the website for the NHS and their guidance about a healthy lifestyle’(15). Another interviewee stated: ‘I know there are resources online, where they have information for each chronic condition… I know there is specific kind of guidance of how to go about giving physical activity education to patients in those areas’ (1).

Moving Medicine (an online PA resource for HCPs) was mentioned specifically by several of the interviewees, with one stating about the moving medicine website: ‘I just have a flick-through on the conditions and then I can either print it out or just made a note on it and then relay that information to the patients’ (1). However, only 33% (5/15) of participants were aware of this resource, and no participants described using it regularly. Reasons for this were described as due to setting and lack of time or familiarity. In contrast, a few interviewees commented on the difficulties of finding the appropriate online or visual resources, with one stating: ‘obviously there are some available but finding the right ones is usually quite tricky’ (3). Only four participants (27%) were familiar with the UK CMO PA guidelines, and participants from either those aware or not aware felt that resources needed more promotion, as most HCPs would not be aware of them.

## 4. Discussion

The aims of this study were to explore the understanding of doctors in the UK of the PA guidelines and barriers and solutions to engaging in discussions on PA with patients. Thoughts and opinions were obtained from doctors in England in a variety of different specialties and at a variety of stages in training. This study found that some issues related to delivering PA advice include a lack of knowledge of the PA guidelines, a lack of priority for giving PA advice, time pressures, and patient engagement. Solutions given included staff training on PA, incorporating PA into the undergraduate curriculum, encouraging staff themselves to be active, providing individualised advice, utilising local exercise services and online or visual resources, and using motivational interviewing to facilitate the advice being given.

### 4.1. Improving Knowledge, Confidence, and the Priority Given to Providing PA Advice

Improving inactivity levels in society is a complex and multifactorial issue, demonstrated well by the fact that limited solutions have been developed to combat the rise in physical inactivity despite extensive evidence of the benefits of being active for health and quality of life. The findings of this study are in line with previous papers highlighting the lack of knowledge and confidence of HCPs in giving PA to patients [10,20,21,22]. Indeed, 10% of junior doctors felt they had been adequately trained in PA [23]. Improving this could lead to PA being discussed during more consultations. Indeed, improved training and education for doctors were highlighted as solutions from the findings of this research.

This study found that PA being an afterthought and not a priority is a reason why it may not be mentioned during consultations. A similar finding of lack of priority to PA advice during consultations was found amongst physiotherapists [22]. Interestingly, one study found that 98.9% of GPs believe that PA is important for health [10]. Therefore, while the benefits are well acknowledged by doctors, it continues to not be thought of as a priority during consultations. Potentially, this is due to PA not being highlighted throughout training as an important aspect to consider. It has been suggested that PA levels be considered a vital sign [24] and, given the wide-ranging impacts of PA on health, it would be prudent for discussion regarding PA to be taught at an early stage of medical training when students are developing and consolidating consultation skills and evaluating vital signs. This may help PA discussions become a natural part of consultations for doctors of the future. The findings of this study, therefore, support previous calls for PA to be better embedded into the training of medical staff and into undergraduate curricula [25,26]. Currently, in the UK, the medical curriculum, which is published by the General Medical Council and forms the framework for all UK medical schools, gives little priority to PA. In fact, there is only one outcome related to PA, and that is within the context of weight loss only [27]. Including PA in curricula at an early stage and throughout training may increase the priority given to discussing PA during consultations. The Office of Health Improvement and Disparities (OHID) currently runs the moving HCPs programme, which is a whole system educational approach to embedding PA into clinical practice [28].

### 4.2. Encouraging a More Physically Active Workforce

Encouraging doctors to be active as a way of encouraging the promotion of PA to patients is an interesting concept and finding of this study. However, this is not a novel concept, with a range of global research suggesting that doctors who are active are more likely to give physical activity advice to their patients [29,30,31]. A study performed in Glasgow in 2019 found that 63.9% of 332 doctors met the recommended volume of aerobic activity, while only 23.5% achieved the recommended account of muscle-strengthening activities [21]. They concluded that their results indicated doctors are as active as the UK general public. Another recent study found that 58% of 245 UK doctors met the PA guidelines, again concluding no significant difference from levels in the general population [31]. Clearly, encouraging medical staff to be active is valuable in promoting health among the staff themselves and the individual benefits they will gain from this and in ensuring a healthy workforce. Previously there have been calls for the NHS to better encourage, support, and facilitate staff to engage in PA [31]. The moral debate of whether doctors need to, or should be, role models for patients regarding health behaviours [32] is beyond the scope of this article. However, the findings of this study support the concept that promoting PA among HCPs might result in PA being discussed more often with patients [33].

### 4.3. Utilising Resources

Time pressures are an issue throughout healthcare and have been highlighted numerous times throughout the literature as a reason why PA is not discussed with patients by HCPs [20,25,34,35]. This is supported by the findings of this study. A solution to this may be optimising the way in which PA can be delivered. Referring to local supported exercise services was highlighted during the interviews.

Utilising online and visual aids to promote PA to patients was also highlighted as a potential solution. Signposting to useful resources may also aid with the issue of having limited time during consultations. In addition, this study found that ensuring PA guidance is individualised is beneficial in optimising the PA advice given. With regard to online and visual aids, the importance of ensuring that they are specific for individual patients has been noted in recent years, with resources such as the CMO’s PA guidance and Moving Medicine (an online aid for providing PA advice to patients) having targeted advice for specific population groups such as age groups and different chronic health conditions [36]. It should be kept in mind that resources that are online may perpetuate inequalities, as patients without the socioeconomic capacity to purchase technology or digital literacy may not be able to easily access these resources.

### 4.4. Limitations and Strengths

There is a risk of participant bias, as those who are more interested in PA may have been more likely to volunteer. However, based on the backgrounds of the doctors interviewed, the research team remained confident that the majority of participants were doctors without a specific PA interest or expertise. Most participants were based in Yorkshire and Humber and northwest England, and it would have been advantageous to gain a broader view from doctors based in other locations in the UK. There is potential for researcher bias and the reliability of interpretations of the interview data, although Braun and Clarke’s six steps were used to reduce this. The data were also analysed independently by DV and KM, and the themes were discussed to reduce the potential for bias. Strengths of this research included in-depth qualitative accounts from doctors about the issues and challenges they faced when promoting PA, as well as possible areas that could be improved to help promote PA. The involvement of this group in shaping interventions is essential, given that HCPs have been identified as being key for promoting PA [3,25].

It should be acknowledged that this project specifically focused on PA guidelines for healthy adults. Alternative guidelines are recommended for specific population groups such as pregnant women, older adults, and those with chronic health conditions. It would be beneficial for future research to explore HCPs’ thoughts and opinions on delivering PA advice for specific groups. It should also be noted that longer durations of higher intensity activity can actually be detrimental to certain health conditions such as chronic fatigue syndrome and long COVID-19 [37].

## 5. Conclusions

This paper provides an updated insight into doctors’ views and opinions on barriers and solutions to discussing PA with patients. It is clear that further work needs to be done to ensure greater awareness of PA guidelines amongst clinicians so that they, in turn, can best advise their patients, improve the health of the nation, and reduce morbidity and mortality. Barriers to delivering PA advice included lack of PA education, time pressures, patient engagement, and limited staff training. Incorporating PA education in undergraduate training was highlighted as a possible solution, as was increasing awareness of resources to support clinicians. Increasing PA advice given to patients requires a multifaceted approach, including increasing education of HCPs, raising awareness of patient resources and awareness of local services, as well as wider healthcare system-wise change.

## Figures and Tables

**Table 1 ijerph-19-12030-t001:** Participant characteristics.

Characteristic	Category	N (%)
**Years of experience**	0–5	3 (20%)
	6–10	11 (73%)
	15+	1 (7%)
**Current primary healthcare setting**	Inpatients	4 (27%)
	Outpatients	2 (13%)
	Inpatients and outpatients	2 (13%)
	Primary Care	6 (40%)
	Academic or leadership role	1 (7%)
**Job role**	Foundation Year 1	3 (20%)
	Medical Registrar	1 (7%)
	Psychiatry Registrar	1 (7%)
	Paediatric registrar	1 (7%)
	General Practice registrar	3 (20%)
	Qualified General Practitioner	3 (20%)
	SEM registrar	2 (13%)
	SEM Consultant	1 (7%)
**UK region**	West Midlands	0
	East Midlands	1 (7%)
	Yorkshire and Humber	4 (27%)
	Northwest	9 (60%)
	London	1 (7%)
**Meeting CMO’s PA guidelines for aerobic exercise**	Yes	12 (80%)
	No	3 (20%)
**Aware of CMO’s PA guidelines**	Yes	4 (27%)
	No	11 (73%)
**Aware of Moving Medicine**	Yes	5 (33%)
	No	9 (67%)

CMO, Chief Medical Officer; N, number; PA, physical activity; SEM, sport and exercise medicine; UK, United Kingdom.

**Table 2 ijerph-19-12030-t002:** Themes and sub-themes.

Theme	Sub-Themes
**Intrinsic factors limiting delivery of PA advice**	Lack of knowledge of PA guidelines
	PA is often an afterthought
**Barriers to delivering PA guidance**	Lack of PA education
	Time pressures
	Patient engagement
**Solutions to increase staff awareness and successful implementation of PA guidelines**	Staff training
	Incorporating PA into undergraduate training
	Encouraging staff to be active themselves
**Methods to optimise PA advice**	Individualised PA advice
	Local exercise services and schemes
	Utilising online & visual resources

## Data Availability

The data presented in this study are available on request from the corresponding author. The data are not publicly available to avoid the potential identification of interviewees based on comments made.

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
