# Peer review of "UK Doctors Delivering Physical Activity Advice: What Are the Challenges and Possible Solutions? A Qualitative Study"

_ijerph, 2022, doi:10.3390/ijerph191912030_

Round 1

Reviewer 1 Report

The paper describes a study that was performed to better understand awareness of physical activity (PA) guidelines among physicians in the UK NHS, and explore perceived barriers and potential solutions to physicians discussing PA with their patients. 

The paper is written in clear and simple language with a good logical flow. 

The Methods are described well and appropriate for the study. 

In the Results section, the themes and subthemes are listed clearly and described with sufficient detail.  I would suggest that the authors please clarify if the order in which the subthemes are listed indicate importance that was ascribed to those subthemes by the participants or if the subthemes are listed in no particular order.

In the Results section, in addition to the participant characteristics presented in Table 1, it would be useful to please report other quantitative-type data collected from the interviews.  This would give the reader a better idea on the background of the participants.  e.g. What percentage of participants said they are meeting PA guidelines? What percentage of participants answered Yes/No for the question on whether or not the 2019 update has been helpful.  What percentage of participants answered Yes/No to having seen the CMO PA infographics? What percentage of participants answered Yes/No for the question on whether or not they knew about Moving Medicine?

The Discussion is well formed, and includes a transparent listing of limitations.  My only suggestion would be to please add a note/warning that the PA recommendations discussed in this article are for healthy adults, and that standard PA recommendations might not be suitable for those who have certain health conditions (e.g., muscular dystrophies, chronic fatigue syndrome, post-acute sequelae of COVID-19, cardiorespiratory insufficiency, etc).  As we know, large blocks of intense PA is harmful for patients with certain health conditions.

The Conclusions are supported by the data.

Author Response

Thank-you for your comments and suggestions all actions and detailed in the attachment,

Author Response

(The authors gave the same response as above.)

Reviewer 3 Report

Although the problem is not new, this manuscript presents a topic which is worth talking about. The authors did an excellent job in interviewing doctors out of various parts of the medicine. Yet, maybe by that, some innovative ideas and novelty is missing.

Below some remarks on the specific parts of the manuscript.

Participants:

More information is needed about the recruitment methods. For example, which channels, organizations and social media platforms were used? Only 15 participants responded, so more information is needed to check for bias (such as the authors also mentioned in the discussion). 

For readers out of the UK the current description of the population is not enough; which type of doctor’s work in the NHS?

à it seems a very heterogenous group of participants (i.e., Table 1 shows a wide range of job settings and roles), why was chosen for this approach? What are the similarities between these doctors to account for the relatively small number of participants? How are differences in characteristics, such as job setting, and aspects as mean patient follow-up duration taken into account when analyzing the data? It may be that doctors who work in primary care feel more urge to start the conversation about PA participation than those who work in an inpatient setting.

Results:

It seems very coincidental that 15 participants, responded, participated and that the exact that same number let to saturation. The authors may report when first signs of saturation were seen and when ‘complete’ saturation was reached.

The quotes miss references. This may help to see differences and similarities between interviewee responses in different themes. For example, the person who responded he/she likes to provide leaflets will not be the same as the person who use motivational interviewing during the conversation. 

Utilising online & visual resources. In this part authors mention Moving Medicine is mentioned specifically. But Moving Medicine is part of a question in the interview guide. It would be interesting to know how many doctors knew about and used the tool? How did the interviewees respond to the suggested guidelines, infographics, and applications available to support the provision of PA advice? (this is mentioned in the conclusion, but not in the result section)

The subtheme motivational interview seems a bit out of place as a separate theme and may fit better in 3.6.1 ‘individual PA advice’ as a preferred way of communication.

Discussion:

Related to the point that PA is often seen as an afterthought, providing medicine than a training about exercise guidelines probably will not help. It would be valid if the authors could reflect on this point somewhat more. Maybe someone else (with perceived benefits by an increased population with an improved lifestyle and/or increased PA participation) need to take over. The authors already mention exercise referral schemes (ERS). Yet, in that part the effect of ERS has been discussed instead of talking about the opportunities to support PA. In addition, the effect discussion better can be removed because the effect of PA advise by a doctor is not discussed on effect either.

The subject mentioned in 4.4 does not refer to any of the results provided in the manuscript.                                         

Conclusion. This conclusion does not match the aim, nor the results provided in the manuscript. When this is the main outcome, the result section should be presented differently.

Minor: reference 37 misses an access date

Author Response

(The authors gave the same response as above.)
